# Elevated Production of Mitochondrial Reactive Oxygen Species via Hyperthermia Enhanced Cytotoxic Effect of Doxorubicin in Human Breast Cancer Cell Lines MDA-MB-453 and MCF-7

**DOI:** 10.3390/ijms21249522

**Published:** 2020-12-15

**Authors:** Azusa Terasaki, Hiromi Kurokawa, Hiromu Ito, Yoshiki Komatsu, Daisuke Matano, Masahiko Terasaki, Hiroko Bando, Hisato Hara, Hirofumi Matsui

**Affiliations:** 1Graduate School of Comprehensive Human Sciences, University of Tsukuba, Ibaraki 305-8577, Japan; azyu0713@gmail.com (A.T.); s0611608@yahoo.co.jp (Y.K.); m.d.d.a.kj@gmail.com (D.M.); 2Department of Breast-Thyroid-Endocrine Surgery, University of Tsukuba Hospital, Ibaraki 305-8577, Japan; 3Faculty of Medicine, University of Tsukuba, Ibaraki 305-8577, Japan; hkurokawa.tt@md.tsukuba.ac.jp (H.K.); k8459616@kadai.jp (H.I.); 4Graduate School of Medical and Dental Sciences, Kagoshima University, Kagoshima 890-0065, Japan; 5Division of Gastroenterology, Faculty of Medicine, University of Tsukuba, Ibaraki 305-8575, Japan; mterasaki@md.tsukuba.ac.jp (M.T.); hmatsui@md.tsukuba.ac.jp (H.M.); 6Division of Breast and Endocrine Surgery, Faculty of Medicine, University of Tsukuba, Ibaraki 305-8577, Japan; harahisa@md.tsukuba.ac.jp

**Keywords:** hyperthermia, reactive oxygen species, ABCG2, doxorubicin

## Abstract

Hyperthermia (HT) treatment is a noninvasive cancer therapy, often used with radiation therapy and chemotherapy. Compared with 37 °C, 42 °C is mild heat stress for cells and produces reactive oxygen species (ROS) from mitochondria. To involve subsequent intracellular accumulation of DOX, we have previously reported that the expression of ATP-binding cassette sub-family G member 2 (ABCG2), an exporter of doxorubicin (DOX), was suppressed by a larger amount of intracellular mitochondrial ROS. We then hypothesized that the additive effect of HT and chemotherapy would be induced by the downregulation of ABCG2 expression via intracellular ROS increase. We used human breast cancer cell lines, MCF-7 and MDA-MB-453, incubated at 37 °C or 42 °C for 1 h to clarify this hypothesis. Intracellular ROS production after HT was detected via electron spin resonance (ESR), and DOX cytotoxicity was calculated. Additionally, ABCG2 expression in whole cells was analyzed using Western blotting. We confirmed that the ESR signal peak with HT became higher than that without HT, indicating that the intracellular ROS level was increased by HT. ABCG2 expression was downregulated by HT, and cells were injured after DOX treatment. DOX cytotoxicity enhancement with HT was considered a result of ABCG2 expression downregulation via the increase of ROS production. HT increased intracellular ROS production and downregulated ABCG2 protein expression, leading to cell damage enhancement via DOX.

## 1. Introduction

The number of cancer patients is increasing worldwide. In 2016, there were 17.2 million cancer cases and 8.9 million deaths globally. Between 2006 and 2016, cancer cases increased by 28% [1]. Among them, breast cancer is the most common incident cancer in women, and morbidity and mortality tend to increase every year [1,2]. Generally, curing a cancer patient who has metastases or recurrence is challenging, and the 10-year survival rate after recurrence is about 5% [3]. The treatment for breast cancer patients with metastases is based on systemic therapy; however, multimodal treatment that combines various therapies, such as radiation therapy and surgery, if it is useful for symptom relief, is conducted sometimes. Hyperthermia (HT) treatment is one of the multimodal therapies that has gained attention in recent years. HT treatment is a noninvasive cancer therapy that, when combined with chemotherapy and radiation therapy, may have an additive effect. HT enhances the effects of cancer treatments, including chemotherapy and radiotherapy, by heating the tumor lesion to 41 °C or more. Heating at 40 °C to 42 °C increases blood flow in the tumor and enhances the hypoxic environment of tumor cells, leading to increased radiation sensitivity, drug uptake into tumor tissue, and sensitivity to anticancer drugs [4,5,6,7]. Recently, it has also been reported that HT affects autoimmunity and makes immunotherapy more effective [8,9]. HT treatment is a promising multidisciplinary treatment for malignant tumors, and clinical studies have reported that a few positive comparative trials have been completed in various cancer types to date, but the mechanisms remain unclear. In April 1990, the Japanese government approved the use of health insurance to cover the costs of electromagnetic hyperthermia combined with radiotherapy. In April 1996, electromagnetic hyperthermia was also approved for clinical uses alone or in combination with chemotherapy or radiotherapy. One of the mechanisms of HT is believed to be related to reactive oxygen species (ROS) released by heat. Additionally, immature cancer blood vessels are easily destroyed by heating, preventing the dissipation of heat, and inducing cytotoxicity. Although the mechanism of HT is not completely elucidated, some studies report that cancer cells released ROS via HT, and these were related to cell damage. We have been studying the intracellular signal pathway by ROS and the relationship between ROS and the ABC transporter, a known cause of anticancer drug resistance. We previously reported that the treatment of gastric cancer with cisplatin elevated intracellular ROS generation and downregulated ABCG2 transporter expression [10]. Similarly, we reported that ABCG2 downregulation by HT-induced elevation of ROS, derived from mitochondria (mitROS), induced an additive effect for photodynamic therapy in gastric cancer cells [11]. The ABCG2 transporter, also known as breast cancer resistance protein (BCRP), is a member of the ABC transporter family that was discovered in human breast cancer cell line MCF-7 and has become resistant to doxorubicin (DOX) [12]. The ABC transporter excretes the drug extracellularly and reduces intracellular accumulation of the drug. This mechanism of drug resistance and the cause of anticancer drug resistance are reported to be the overexpression of ABC transporters [13]. DOX is excreted from ABCG2 transporters. Thus, we hypothesized that HT-induced elevation of intracellular mitROS might have an additive effect with intracellular DOX accumulation by the downregulation of ABCG2 transporter expression. Additionally, four subtypes of breast cancers are classified by the presence or absence of hormone receptors and HER2 receptors, and treatment varies slightly by subtype. The most common subtype is estrogen receptor-positive breast cancer, and HER2-positive breast cancer is known to have high growing ability. To date, few studies have examined HT in breast cancers by subtype, and it remains unclear for which subtype is HT more effective. In the present study, we investigated DOX cytotoxicity on breast cancer cells by adding DOX to HT, using two types of human breast cancer cells with different subtypes.

## 2. Results

### 2.1. Hyperthermia Enhanced DOX Cytotoxicity

DOX was measured via a WST assay, which is a cell viability assay that uses the Cell Counting Kit-8. Figure 1 shows cell viability after combination treatment with HT and DOX. Cell viability of MDA-MB-453 and MCF-7 after combination treatment decreased significantly in 1 and 10 µM DOX-treated groups, compared with that in the non-HT group that had only DOX treatment. Furthermore, cell viability of both cell lines decreased dose-dependently of DOX. These results indicated that the combination treatment of HT and DOX showed an additive effect.

### 2.2. Mitochondrial ROS Increased by HT

Using ESR, ROS production levels in living cells with or without HT treatment were evaluated. Figure 2a,c shows ESR spectra after HT and non-HT treatment of MDA-MB-453 and MCF-7 cells, and Figure 2b,d shows the calculation of the ratio of signal-to-noise peak intensities. The ESR spectra and the signal intensity in MDA-MB-453 and MCF-7 cells with HT increased significantly compared with those in cells without HT treatment. Mitochondrial ROS of MCF-7 and MDA-MB-453 cells after HT or non-HT treatment were measured using MitoSOX^TM^, which is a mitochondrial superoxide indicator. In Figure 3a, fluorescence intensity after HT increased, compared with that in the non-HT groups, and when quantified by ImageJ Fiji, the difference was significant for both MDA-MB-453 and MCF-7 (Figure 3b). These results indicated that HT enhances intracellular ROS production, especially in mitochondria.

### 2.3. Hyperthermia Downregulated the Expression of ABCG2 Transporter

The expressions of intracellular ABCG2 transporters in cells with or without HT treatment were analyzed using Western blotting. Figure 4 shows ABCG2 expressions 24 h after HT or non-HT treatment. Compared with the non-HT groups, the expression of ABCG2 transporter significantly decreased in the HT group of MDA-MB-453. In MCF-7, the expression of ABCG2 transporter in the HT group decreased, but not significantly. These results showed that HT downregulated ABCG2 expression more readily in MDA-MB-453 than in MCF-7.

## 3. Discussion

In this study, we examined the additive effects of HT on ROS production and DOX using two different human breast cancer cell lines.

Cell viability assay after combination treatment with DOX and HT showed that cell viability was significantly decreased in both MDA-MB-453 and MCF-7, compared with that in the non-HT groups. These results suggested that HT enhances the effect of DOX. Further, no difference in cell viability was observed without DOX, indicating that HT at 42 °C for 1 h was safe for cells (Figure 1). This additive effect might be involved with overproduction of mitROS by HT and reduced expression of the ABCG2 transporter.

HT is known to increase intracellular ROS [14,15]. In the present study, the ratio of signal-to-noise peak intensities of ESR was also significantly increased by HT at 42 °C for 1 h, compared with those in the non-HT groups (Figure 2). Similarly, in experiments that use MitoSOX^TM^ to detect mitROS, fluorescence intensities were significantly enhanced in the HT group (Figure 3). These results indicated that the ROS detected by ESR were of mitochondrial origin.

Although the mechanism of intracellular signal transduction by ROS from mitochondria remains unclear, we previously reported that mitROS increase was associated with ABCG2 transporter downregulation [10,11]. Again, as shown in Figure 4, the expression of ABCG2 transporter significantly decreased with HT in MDA-MB-453 after 24 h. There was a decreasing trend in MCF-7, although not significant. This lack of difference might be caused by the difference in ROS production between MDA-MB-453 and MCF-7 after HT. This was shown by the ratio of ESR spectra and signal-to-noise peak intensities in Figure 2, depicting that MDA-MB-453 produced more ROS than MCF-7 did. The fluorescence image and its quantification graph in Figure 3 also showed that MCF-7 produced less ROS than MDA-MB-453 did. In summary, our results suggested that downregulation of ABCG2 transporter expression was related to the production of mitROS.

The overexpression of ABCG2 transporter, which is one of the ABC transporters, also called BCRP, is known to cause drug resistance in breast cancers [12,13]. Among the ABC transporters, ABCB1, MDR1, ABCB1 (MRP1), and ABCG2 (BCRP) are responsible for extracellular excretion of various anticancer drugs. Therefore, the overexpression of these transporters in cancer cells indicates anticancer drug resistance, and reduced expression in normal cells enhances anticancer drug toxicity as a side effect. This can lead to an increase in adverse events. Therefore, specific control of ABCG2 transporter expression in cancer cells alone would enhance the effects of anticancer drugs without increasing adverse events. In this study, we used only human breast cancer cell lines and did not compare them with normal cells; hence, we cannot examine whether the results were cancer cell-specific or not. However, it has been reported in various studies that mitROS production is specifically increased in cancer cells, compared with that in normal cells [16,17], and our previous studies have also reported a cancer cell-specific increase in mitROS between normal and cancer cells [18,19]. We also reported that increased mitROS was associated with downregulation of ABCG2 [10,11]. Thus, we expect HT to increase mitROS and decrease the expression of ABCG2 transporters in cancer cells in a specific manner, regardless of cancer type. Additionally, body surface malignancies such as breast cancer can be treated with local HT only. With local HT only, one can expect an increase in ROS in cancer cells in a specific manner. This phenomenon would not lead to an increase in adverse events and would be expected to enhance the effect of anticancer drugs. Some studies have reported that local HT using magnetic nanoparticles (MNPs) was effective for cancer treatment. Piehler, S. et al. (2020) have reported that local magnetic HT using DOX-functionalized MNPs (43 °C, 1 h) was effective for tumor regression in breast cancer cells (BT474) in vivo [20]. Thus, local HT combined with chemotherapy may be further developed in the future.

The mechanism of HT remains unclear, and several studies have been conducted to date, showing the following biological benefits [6,21,22]: (1) HT at 42 °C–43 °C decreases cell viability with heating time, which is common in many cells, and susceptibility is dependent on tissue type; (2) tumor tissues are more susceptible to heating than normal tissues; (3) hypoxic cells are more sensitive to HT than aerobic cells; (4) cell recovery from radiation or anticancer drugs is prevented by HT; (5) the sensitivity of the cell cycle to radiation and HT is different. Based on this biological evidence, heating devices have been developed, and several clinical trials have shown efficacy. Combination treatment of HT and chemotherapy is known to be most effective when conducted concurrently. However, conducting HT concurrently with chemotherapy is currently challenging because of problems such as space and equipment. Consequently, HT tends to be avoided due to operational inconveniences and poor cost–benefit characteristics.

We acknowledge there are several limitations in this study. First, this study is in vitro and does not include in vivo results. In addition, this was a little difficult experiment to reproduce to achieve the same results. The Appendix A shows the result of additional experiments in cell viability assay after combination treatment with higher concentration of DOX (Appendix A). The trend of Appendix A was similar to Figure 1, but not exactly the same as in Figure 1. However, in both cell lines, significant differences were also observed in some groups and this result also indicates an additive effect of combination treatment with HT and chemotherapy. Therefore, the combination treatment with HT and DOX would be a promising treatment.

In conclusion, we demonstrated the combination efficacy of HT with an anticancer drug after 24 h, albeit in vitro, and showed that HT enhanced the effect of the anticancer drug via overproduction of ROS, followed by downregulation of a drug-resistant transporter. The present study could be a major step forward in the future prospects of HT. We would like to continue our in vivo studies and further develop our results as a form of clinical application.

## 4. Materials and Methods

### 4.1. Cell Culture and Hyperthermia Treatment

The estrogen receptor (ER)-positive human breast cancer line MCF-7 (RCB1904) and the ER-negative human breast cancer cell line MDA-MB-453 (RCB1192) were purchased from the RIKEN cell bank. MCF-7 cells were cultured at 37 °C with 5% CO_2_ in E-MEM with L-glutamine and phenol red (Wako Pure Chemical Industries, Ltd., Osaka, Japan) containing 10% fetal bovine serum (GE Healthcare Life Sciences, Inc., Logan, UT, USA), 1% penicillin/streptomycin (Wako Pure Chemical Industries, Ltd.), 1.0 mM sodium pyruvate solution (Wako Pure Chemical Industries, Ltd.), and 1% MEM non-essential amino acid solution (Thermo Fisher Scientific, Inc., Waltham, MA, USA). MDA-MB-453 cells were cultured at 37 °C without CO_2_ in Leibovitz′s L-15 Medium with L-glutamine, phenol red, and sodium pyruvate (Wako Pure Chemical Industries, Ltd.) containing 10% fetal bovine serum (GE Healthcare Life Sciences, Inc.) and 1% penicillin/streptomycin (Wako Pure Chemical Industries, Ltd.).

The hyperthermia treatment was based on our previous study [11]. The MCF-7 cells were incubated at 42 °C with 5% CO_2_ for 1 h using a standard incubator, and the MDA-MB-453 cells were incubated in the same way, but without CO_2_.

### 4.2. Cell Viability Assay after Treatment in Combination with HT and DOX

The viability of the cells was calculated using a Cell Counting Kit-8 (DOJINDO LABORATORIES, Kumamoto, Japan) in accordance with the manufacturer′s protocol. MCF-7 and MDA-MB-453 cells were cultured on 96-well plates at 5.0 × 10^3^ cells/well and incubated overnight. The cells were treated at 42 °C for 1 h and then incubated at 37 °C for 24 h. Thereafter, the culture supernatant was aspirated, and the medium was replaced with a fresh medium containing 0, 0.1, and 1 µM DOX (Wako Pure Chemical Industries, Ltd.). Then, the cells were incubated at 37 °C for 24 h. After incubation, the medium was replaced with a fresh medium containing 10% Cell Counting Kit-8, and the cells were further incubated for 1 h. The absorbance at 450 nm was measured using a DTX880 multimode microplate reader (Beckman Coulter, Inc., Brea, CA, USA).

### 4.3. Electron Spin Resonance Measurement

Intracellular ROS production levels in HT-treated MCF-7 and MDA-MB-453 cells were measured using electron spin resonance (ESR) in accordance with the protocol described in a previous report [22]. The cells were seeded on cover slides (49 × 5 × 0.2 mm) and incubated overnight. The cells were incubated at 37 °C or treated at 42 °C for 1 h, and the slides were then immersed in a respiratory solution containing 5 mM succinate (Sigma-Aldrich Co., St. Louis, MO, USA), glutamate (Sigma-Aldrich Co.), malate (Wako Pure Chemical Industries, Ltd.), nicotinamide adenine dinucleotide (NADH) (Sigma-Aldrich Co.), and a 5 μL 5,5-dimethyl-1-pyrroline-*N*-oxide (DMPO) solution (DOJINDO LABORATORIES). The cell-attached glass cover slide was placed on a tissue glass and inserted into the ESR apparatus. All ESR spectra were obtained using a JEOL-TE Xband spectrometer (JEOL Ltd., Tokyo, Japan) under the following conditions: 7.5 mT sweep width, 1000 gain, 0.1 mT modulation width, and 10 mW incident microwave power. ESR spectra data were analyzed using a Win-Rad Radical Analyzer System (Radical Research Co., Ltd., Tokyo, Japan).

### 4.4. Detection of Mitochondrial ROS

Mitochondrial ROS production after HT treatment was estimated using a MitoSOX^TM^ Red superoxide indicator (Thermo Fisher Scientific, Inc.). MCF-7 and MDA-MB-453 cells were seeded on a 96-well cell culture plate at a density of 1.0 × 10^4^ cells/well and incubated overnight. The cells were incubated at 37 °C or 42 °C for 1 h, and the medium was then replaced with an MSF solution (Modified HBSS; 5.4 mM KCl, 136.9 mM NaCl, 8.3 mM Glucose, 0.44 mM KH_2_PO_4_, 0.33 mM Na_2_HPO_4_ 7H_2_O, 10.1 mM HEPES, 1.0 mM MgCl_2_ 6H_2_O, and 1.0 mM CaCl_2_ 2H_2_O) including 5 μM MitoSOX^TM^. The cells were incubated at 37 °C for 30 min. The fluorescence images of MitoSOX^TM^ were captured using an epifluorescence microscope (BZ-X710, Keyence Co., Osaka, Japan), and, using the image processing software package ImageJ Fiji (NIH, Bethesda, MD, USA), the fluorescence intensities of MitoSOX^TM^ were analyzed.

### 4.5. Western Blotting

ABCG2 expression was examined using Western blotting to examine the relationship between cellular ROS production and ABCG2 protein expression levels after HT treatment. MCF-7 and MDA-MB-453 cells were incubated overnight in 100 mm dishes. Afterwards, the cells were incubated at 37 °C or 42 °C for 1 h and then incubated at 37 °C for 24 h. Whole-cell lysates were prepared by rinsing the cells three times with PBS and adding a radioimmunoprecipitation assay buffer containing a protease inhibitor cocktail (Thermo Fisher Scientific, Inc.) on ice, then treated with a NuPAGE^®^ LDS Sample Buffer (Thermo Fisher Scientific, Inc.); afterwards, the samples were heated at 95 °C for five minutes. For sodium dodecyl sulfate–polyacrylamide gel electrophoresis, the cell lysates were added into wells of NuPAGE^®^ Novex^®^ 4–12% Bis-Tris protein gels (Thermo Fisher Scientific, Inc.). The sample proteins were electrophoresed at 100 V for 60 min and then transferred to polyvinylidene difluoride (PVDF) membranes (Bio-Rad Laboratories, Hercules, CA, USA) via electrophoresis at 1.2 mA/cm^2^ for 60 min. The membranes were blocked for 60 min with a PVDF blocking reagent from Can Get Signal^®^ (TOYOBO Co. LTD., Osaka, Japan) and then incubated at 4 °C overnight with anti-rabbit ABCG2 antibody (Cell Signaling Technology, Inc., Danvers, MA) that was diluted at 1:1000 in Can Get Signal Immunoreaction Enhancer Solution 1 (TOYOBO Co., LTD.). After incubation with primary antibodies, the membranes were washed three times with PBS containing 0.1% (*v/v*) Tween 20 (Sigma-Aldrich Co.) (PBS-T) for 10 min and incubated with horseradish peroxidase-conjugated anti-rabbit IgG secondary antibody (Cell Signaling Technology, Inc.) diluted in Can Get Signal Immunoreaction Enhancer Solution 2 (TOYOBO Co., LTD.) at 1:1000 for 60 min. The secondary antibody solution was removed, and the membranes were washed three times with PBS-T. The Lumina forte Western HRP substrate (Millipore Co., Billerica, MA) was applied to the membranes. Images of the blots were captured on an ImageQuant LAS4000 (GE Health Care Japan, Tokyo, Japan). As sample loading control, anti-β-actin antibody (Cell Signaling Technology, Inc.) was used to detect the β-actin protein.

### 4.6. Statistical Analysis

Statistical analysis was conducted using SPSS Statistics 24 (International Business Machines Corporation, Armonk, NY, USA). Data were expressed as mean ± S.D., and individual groups were compared using Student’s *T*-test; *p* < 0.05 was considered as statistically significant.

## Figures and Tables

**Figure 1 ijms-21-09522-f001:**
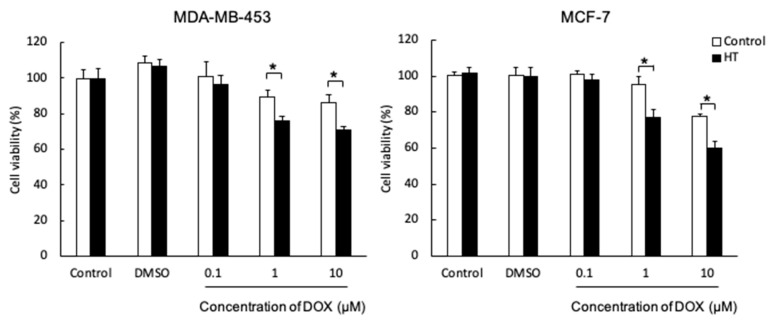
MDA-MB-453 and MCF-7 cells were treated with or without hyperthermia (HT) and cell viabilities after doxorubicin (DOX) treatment were measured using the CCK-8 method. Statistical significance was tested using Student’s *t*-test. *n* = 10 (MDA-MB-453), *n* = 4 (MCF-7), error bar; S.D. * *p* < 0.01.

**Figure 2 ijms-21-09522-f002:**
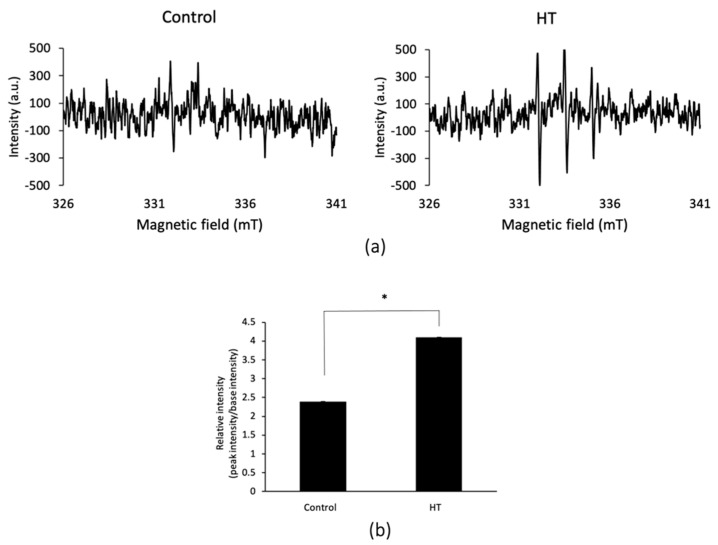
Intracellular reactive oxygen species (ROS) production levels after HT or non-HT treatment were measured using electron spin resonance (ESR). Representative ESR signal waves in MDA-MB-453 (**a**) and MCF-7 (**c**) are shown. Signal-to-noise ratio in MDA-MB-453 (**b**) and MCF-7 (**d**) are represented in graphs. Statistical significance was tested via Student′s *t*-test. *n* = 3 (MDA-MB-453), *n* = 4 (MCF-7), error bar; S.D. * *p* < 0.01.

**Figure 3 ijms-21-09522-f003:**
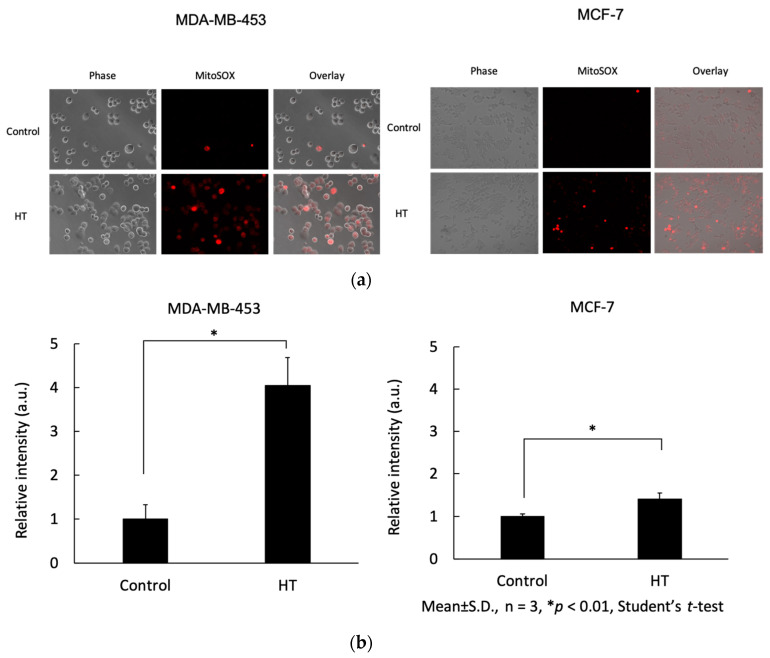
Mitochondrial ROS after HT or non-HT treatment in MCF-7 and MDA-MB-453 cells were measured using MitoSOX^TM^. (**a**) The fluorescence images were captured using an epifluorescence microscope. (**b**) The fluorescence intensities were analyzed using ImageJ Fiji. Statistical significance was tested via Student′s *t*-test. *n* = 3, error bar; S.D. * *p* < 0.01.

**Figure 4 ijms-21-09522-f004:**
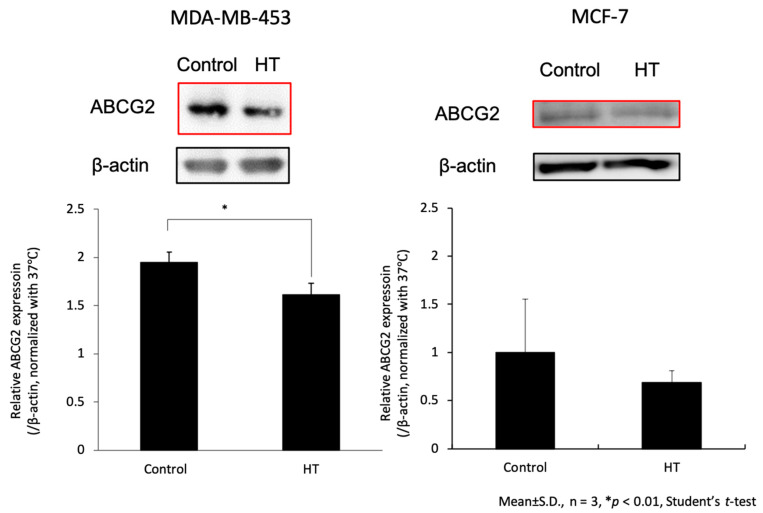
ABCG2 protein expression transitions after HT or non-HT treatment were analyzed using Western blotting. Statistical significance was tested via Student′s *t*-test. *n* = 3, error bar; S.D. * *p* < 0.01.

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
