# Peer review of "Elevated Production of Mitochondrial Reactive Oxygen Species via Hyperthermia Enhanced Cytotoxic Effect of Doxorubicin in Human Breast Cancer Cell Lines MDA-MB-453 and MCF-7"

_ijms, 2020, doi:10.3390/ijms21249522_

Round 1

Reviewer 1 Report

In their study, the Authors estimated the effects of hyperthermia on the sensitivity of two breast cancer cell lines to doxorubicin. Furthermore, they correlated those effects directly to ROS production and ABCG2 transporter expression.

For this reason, the story is promising, because it can give new information on the relationship between HT and cellular sensitivity to an anticancer drug. However, the paper is rather inconclusive and provides only partial data on the inhibitory effects of HT. Therefore, I cannot recommend it for publication in its present form. Below, I enclose several comments that may help to increase the impact of the paper:

Major points:

The effect of the association of DOX and hyperthermia shown in Figure 1 gives not striking evidences on the synergistic/additive (?) effects on cell viability (Line 207). Authors should present the data as semi-Log DOX Dose/response +/- HT curves by showing complete gradient doses that covers the DOX IC50 for both cell lines at given incubation time, from ~100% up to ~0% cell viability. In this way it would be possible to fully appreciate the curve differences, and if the contribution of HT is significant. Furthermore, this approach could give reliable insight on the correlation occurring with the data observed on ESR, mitochondrial ROS and ABCG2 transporter expression.

Data, as presented, seem in contrast with the conclusions. In fact, the DOX +/- HT effects are more evident on MCF-7 viability (although it is hard to state since the IC50s are not presented), while the ESR, mitochondrial ROS and ABCG2 transporter expression data are evident only for MDA-MB-453, where low synergy/additive effects were demonstrated on cell viability.

Authors state at Line 198 that “These results showed that HT downregulated the expression of ABCG2 more readily in MDA-MB-453 than in MCF-7.” How do they try to explain the molecular mechanism of such a difference? Is the hormone sensitivity involed?.....

The use of the same cell lines made DOX-resistant could support the experimental plan and strengthen the importance of HT.

At Line 223 the sentence “This is shown by the ratio of ESR spectra and signal-to-noise peak intensities in Fig. 2, depicting that MCF-7 produced more(?) ROS than did MDA-MB-453. Probably the correct sentence is “This is shown by the ratio of ESR spectra and signal-to-noise peak intensities in Fig. 2, depicting that MCF-7 produced less(!)  ROS than did MDA-MB-453.

Statistical analysis - In section Materials and methods Authors wrote about using two tests (t-test and ANOVA), but Figures and data are always described only by t-test.

Authors do not report which ANOVA have adopted to analyze their data (one-way?) or if they used any multicomparisons tests further analyzed by post-test (e.g. Dunnet, Tukey's,..).The data should be presented on Figure legends as it was analyzed by statistical tests, results of ANOVA should also be provided for each graph, especially for MCF-7 data presented on Figure 2 panel c and d, and in Figure 3 panel b.

The Authors stated that they cultured MDA-MB-453 cells at 37°C without CO2 in Leibovitz’s L-15 medium (Line 86). It is not clear how they made this step and why. Can the Authors exclude that the different CO2 % did not influence the response to heat insult and that it is not responsible for the differences observed by comparing the two cell lines?

There are not enough data on the hyperthermia procedure. As it is described, it is not possible to fully appreciate the accuracy of the procedure. E.g.: was this procedure performed by incubating the cells with 5% C02 (standard incubator)?.

Minor points:

Please improve the quality of Figure 3 – MCF-7 panel (top right)

Please improve the readability of the text by correcting grammar errors along the paper (e.g. Line 127:  To study ABCG2 expression was examined using western blotting to examine the…..; Line 267: In To conclude, we….) or adding commas where needed (e.g. Line 130 to 133)

Please uniform the reference number format into the Introduction section (ref. 10 and 11) and Line 247

Author Response

Dear Reviewer

I apologize for the delay in replying to you.

We wish to express our strong appreciation to the reviewers for their insightful comments on our paper. We feel the comments have helped us significantly improve the paper.

Please check the new file I uploaded.

Thank you so much   Sincerely, Azusa Terasaki, M.D.   Graduate School of Comprehensive Human Sciences,  University of Tsukuba  1-1-1 Tennodai, Tsukuba, Ibaraki, Japan TEL: +81-29-853-3466 (ex. 3466)
E-mail: azyu0713@gmail.com

Reviewer 2 Report

The publication describes the studies on enhancing the DOX cytotoxic effect in breast cancer through hyperthermia. I found such publications in the Web of Science 216 database, e.g. recently Nanomaterials 2020, 10, 1016. The authors do not refer to these publications in their paper. The revised manuscript should include an overview of these publications and a discussion of the proposed mechanism. Also in the discussion, the authors should compare their results with the literature data.

Other minor remarks:
- In my opinion, generally known sentences from the Introduction should be deleted, lines 34-39.
 Magnetic hyperthermia should be used instead of electromagnetic hyperthermia.
- The authors should explain why the estrogen receptor cell line was selected to the studies.
- There is no synergistic effect, since hyperthermia itself has no effect. Better is write about enhancing the cytotoxic effect of DOX by increasing the temperature.

Author Response

Dear Reviewer

I apologize for the delay in replying to you.

We wish to express our strong appreciation to the reviewers for their insightful comments on our paper. We feel the comments have helped us significantly improve the paper.

Please check the new files I uploaded.

Thank you so much

Sincerely,
Azusa Terasaki, M.D.

Graduate School of Comprehensive Human Sciences, 
University of Tsukuba 
1-1-1 Tennodai, Tsukuba, Ibaraki, Japan
TEL: +81-29-853-3466 (ex. 3466)
E-mail: azyu0713@gmail.com

Round 2

Reviewer 1 Report

The manuscript was improved by answering to most of the major points. Some key points as IC50 calculation and different Figure 1 presentation would strength the experimental observations.

Author Response

Thank you for your kind comments.

Reviewer 2 Report

Now the paper can be accepted.

Author Response

Thank you for your kind comments.